# An inhibitory corticostriatal pathway

Crystal Rock, Hector Zurita, Charles Wilson, Alfonso junior Apicella*

Department of Biology, Neurosciences Institute, University of Texas at San Antonio, San Antonio, United States

**Abstract** Anatomical and physiological studies have led to the assumption that the dorsal striatum receives exclusively excitatory afferents from the cortex. Here we test the hypothesis that the dorsal striatum receives also GABAergic projections from the cortex. We addressed this fundamental question by taking advantage of optogenetics and directly examining the functional effects of cortical GABAergic inputs to spiny projection neurons (SPNs) of the mouse auditory and motor cortex. We found that the cortex, via corticostriatal somatostatin neurons (CS-SOM), has a direct inhibitory influence on the output of the striatum SPNs. Our results describe a corticostriatal long-range inhibitory circuit (CS-SOM inhibitory projections → striatal SPNs) underlying the control of spike timing/generation in SPNs and attributes a specific function to a genetically defined type of cortical interneuron in corticostriatal communication.

## Introduction

It is very well established that cortical neurons regulate the activity of spiny projection neurons (SPNs) in the striatum through long-range glutamatergic/excitatory projections (*Landry et al., 1984*; *Wilson, 1987*; *2004*; *Graybiel et al., 1994*; *Lovinger and Tyler, 1996*; *Reiner et al., 2003*; *Kress et al., 2013*), while inhibition is mediated by local feed-forward and feed-back circuits (for review [*Tepper et al., 2008*]). The feed-forward circuit is characterized by GABAergic striatal interneurons that receive excitatory inputs from the cortex and monosynaptically inhibit SPNs. The feedback circuit is characterized by SPNs and their interconnections via local axon collaterals (*Calabresi et al., 1991*; *Kawaguchi, 1993*; *Kita, 1993*; *1996*; *Kawaguchi et al., 1995*; *Mallet et al., 2005*; *Planert et al., 2010*; *Ibanez-Sandoval et al., 2011*). Because striatal neuronal activity has been shown to be involved in movement, learning, and goal-directed behavior (*Graybiel, 1995*; *Schultz et al., 2003*; *Barnes et al., 2005*; *Kreitzer and Malenka, 2008*), it is crucial to understand the cortical connectivity pattern and dynamics that shape the flow of information in the striatum.

Anatomical studies using retrograde tracers and immunohistochemistry have proposed that between 1–10% of the GABAergic 'interneurons' in rodents, cats, and monkeys also give rise to long-range corticocortical projections (*McDonald and Burkhalter, 1993*; *Tomioka et al., 2005*; *Higo et al., 2007*; *Tomioka and Rockland, 2007*; *Higo et al., 2009*). A growing body of evidence suggests that many of these projections arise from somatostatin-expressing neurons (*Tomioka et al., 2005*; *Higo et al., 2007*; *Tomioka and Rockland, 2007*; *Higo et al., 2009*; *McDonald et al., 2012*; *Melzer et al., 2012*). A previous study has demonstrated the presence of a corticostriatal GABAergic projection (*Tomioka et al., 2015*), but the cells of origin and physiological function of this GABAergic projection from the cortex to the dorsal striatum were not explored.

To test the hypothesis that the cortex has a direct inhibitory influence on the output neurons of the striatum, we measured the response of SPNs to optogenetic activation of corticostriatal somatostatin neuron (CS-SOM) axons. Our results describe a previously unknown corticostriatal direct inhibitory circuit (**CS-SOM inhibitory projections → striatal SPNs**) underlying the control of spike timing/generation in SPNs and attribute a specific function to a genetically defined type of cortical interneuron in corticostriatal communication. Overall this suggests that the timing and ratio of cortical

*For correspondence: alfonso. apicella@utsa.edu

**Competing interests:** The authors declare that no competing interests exist.

**eLife digest** The striatum is located beneath the cerebral cortex, where it contributes to processes including learning and movement. The Spanish anatomist Ramon y Cajal, working in the early 20[th] century, was the first to observe individual neurons extending from the cortex to the striatum. Cajal published drawings of these neurons in his now celebrated anatomical papers, but knew little about their properties.

In the 1980s, advances in techniques for labeling individual cells made it possible to study these neurons in detail. The results suggested that the pathways are exclusively excitatory: that is, the cortical neurons always increase the activity of their partners in the striatum. However, this result made it difficult to explain why electrically stimulating the cortex can sometimes reduce or inhibit the activity of the striatum. To reconcile these facts, most people assumed that inhibition must occur when excitatory cortical neurons activate networks of inhibitory cells within the striatum itself.

Rock et al. now challenge this view by providing anatomical and physiological evidence for the existence of long-range inhibitory pathways from the cortex to the striatum in the mouse brain. These inhibitory neurons project from the auditory and motor regions of the cortex, and contain a substance called somatostatin. These neurons form connections with a specific type of striatal neuron called medium spiny neurons, which in turn project to other brain regions outside the striatum. The inhibitory cortical neurons can alter the activity of the medium spiny neurons, and can therefore directly control the output of the striatum.

The discovery that the striatum receives both excitatory and inhibitory inputs from cortex suggests that the timing and relative strength of these inputs can affect the activity of the striatum. Future experiments should examine whether this is a general mechanism by which sensory stimuli can influence the processes controlled by the striatum, such as movement.

excitatory and inhibitory inputs to the dorsal striatum, by shaping the activity pattern of SPNs, determines behavioral outcomes.

## Results

### Anatomical and electrophysiological properties of corticostriatal somatostatin neurons

To visualize long-range GABAergic projections originating in the cortex and terminating in the dorsal striatum, we conditionally expressed GFP in somatostatin-expressing (SOM) interneurons by injecting AAV.GFP.Flex into the right auditory cortex (AC) of SOM-Cre-tdTomato transgenic mice (*Figure 1a*). GFP was colocalized with SOM/tdTomato-expressing neurons in the AC (*Figure 1b* left) and GFP-positive axons were visible in coronal sections of the right dorsal striatum in these mice (*Figure 1b* middle and right).

Next, to determine the layer of origin for the long-range corticostriatal SOM neurons (from this point forward referred to as CS-SOM neurons), we injected the AAV.GFP.Flex virus into the right dorsal striatum of SOM-Cre transgenic mice (*Taniguchi et al., 2011*) (*Figure 1c*). Using this virus, CS-SOM neurons were retrogradely labelled with GFP, and their somata were present primarily in layers 5 and 6 of the AC (*Figure 1d*, middle). This approach allowed us to visually identify and record from layer 5 and 6 CS-SOM neurons using whole-cell patch clamp. Confocal images of biocytin-filled CS-SOM neurons showed that they are similar to SOM interneurons in their morphology and send an axonal projection into the subcortical white matter (*Figure 1d*, right; Neurolucida-reconstructed CS-SOM neuron *Figure 1e*). We verified their identity based on the comparison with electrophysiological properties of SOM interneurons (*Ma et al., 2006*). These properties include a wide action potential and low rheobase (the smallest current step evoking an action potential) (*Figure 1f*, top; action potentials in CS-SOM neurons, shown in black, are wider than those from fast-spiking interneurons, example in red). The responses to current steps in CS-SOM neurons were typical for SOM interneurons (*Figure 1f*, bottom; notice the sag from a hyperpolarizing current step). Basic electrophysiological properties for CS-SOM neurons (*n* = 13) included (*Figure 1g*): resting membrane

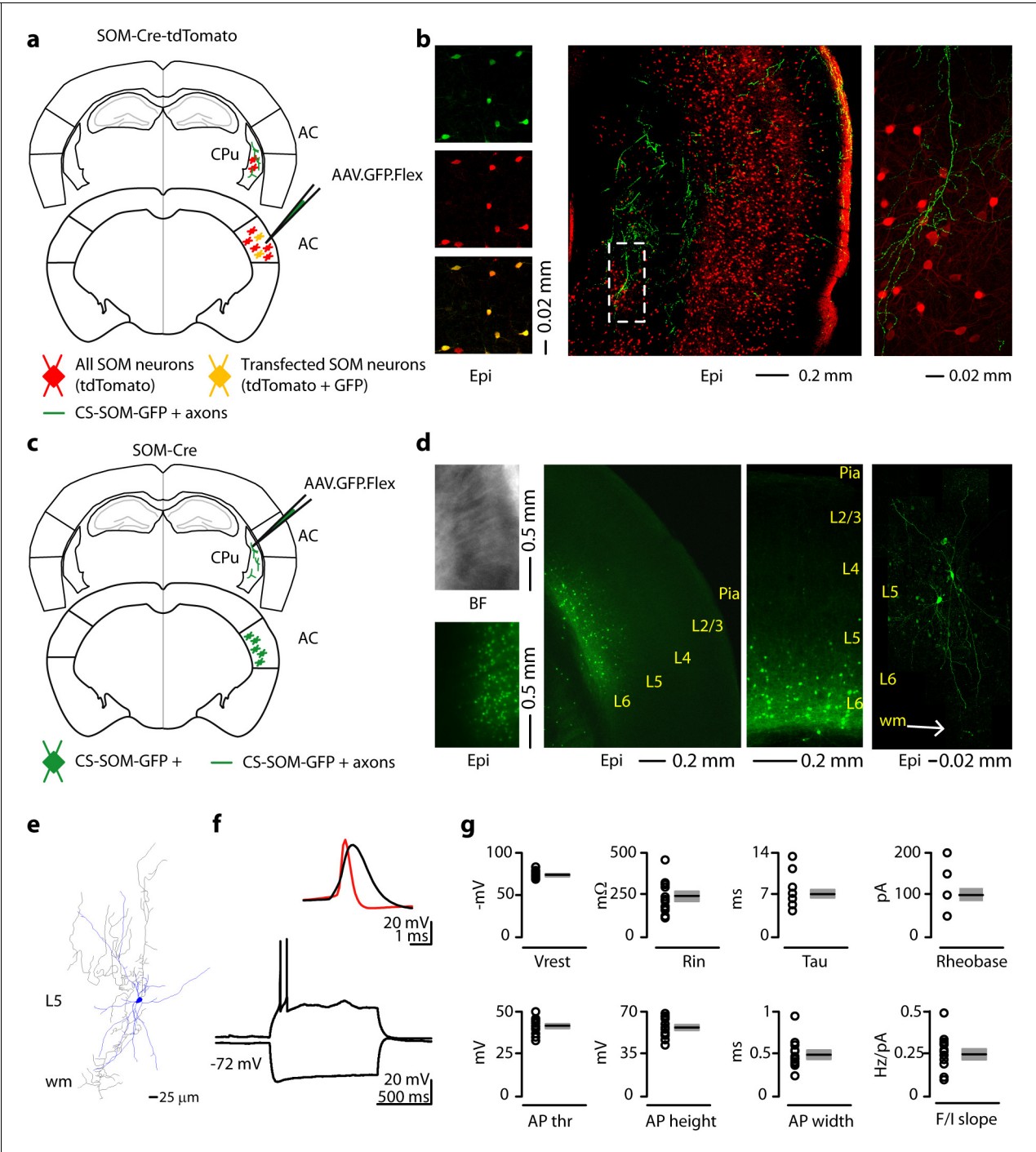

**Figure 1.** Morphological characteristics, axonal projections and electrical properties of long-range CS-SOM neurons in the mouse auditory cortex. (**a**) Schematic depicting injection site using the SOM-Cre-tdTomato transgenic mouse line to identify CS-SOM neurons and their projections to the dorsal striatum. Bottom, auditory cortex: AAV.GFP.Flex injection site; yellow CS-SOM somata coexpressing GFP and tdTomato. Top, dorsal striatum: green CS-SOM GFP-positive axons; red SOM tdTomato-positive interneurons. (**b**) Epifluorescence images of SOM GFP-positive neurons. Top, left: GFP-positive SOM neurons in the auditory cortex identified by viral injection of AAV.GFP.Flex in the SOM-Cre-tdTomato transgenic mouse line. Middle, left: tdTomato-expressing SOM neurons in the SOM-Cre-tdTomato transgenic mouse line. Bottom, left: overlay of GFP and tdTomato images. Middle, the dashed box indicates the location GFP-positive axons from CS-SOM neurons in the dorsal striatum and the location of image in the right panel. Right, higher magnification of GFP fluorescence of CS-SOM axons in the dorsal striatum. (**c**) Schematic depicting injection site using the SOM-Cre transgenic mouse line to identify CS-SOM neurons by anatomical retrograde transfection. Top, striatum: AAV.GFP.Flex injection site. Bottom, auditory cortex: green CS-SOM GFP positive somata. (**d**) Bright-field (top left) and epifluorescence (bottom left) images of striatal SOM interneurons transfected with AAV.GFP.Flex. Middle (left and right), epifluorescence image of laminar distribution of CS-SOM neurons in the auditory cortex identified by anatomical

*Figure 1 continued on next page*

*Figure 1 continued*

retrograde transfection. Right, high-resolution image of a biocytin-labeled CS-SOM neuron. (**e**) Morphological reconstruction of one CS-SOM neuron (dendrites, blue; axons, gray). (**f**) Bottom, train of action potentials recorded in a GFP-positive CS-SOM neuron during step current injection (1.0 s, 100 pA pulse). Top, single action potential from GFP-positive CS-SOM neuron (black); compare to an action potential from a fast-spiking interneuron (red). (**g**) Summary plot of $V_{rest}$: resting membrane potential; $R_i$: input resistance; Tau: membrane time constant; Rheobase, the smallest current step evoking an action potential; AP thr: action potential threshold; AP height: action potential height; AP half-width: action potential half-width; and *F/I slope* from CS-SOM neurons (*n* = 13), including group averages (± s.e.m.).

potential, −72.93 ± 1.16 mV; input resistance, 240 ± 26.98 MΩ; membrane time constant, 6.84 ± 0.71 ms; rheobase, 100 ± 15pA; action potential threshold, −41.35 ± 1.33 mV; action potential height, 54.18 ± 2.21 mV; action potential width, 0.49 ± 0.05 ms; F-I slope, 0.24 ± 0.03 Hz/pA step.

Although, in our experience, AAV1.Flex viral vectors (*Atasoy et al., 2008*) exhibited both anterograde and retrograde (*Rothermel et al., 2013*) transfection capabilities, when we injected this virus in the cortex we only observed anterograde labeling of CS-SOM neurons (i.e. no SOM somata transfected in the dorsal striatum; see *Figure 1b*, right). In contrast, when we injected this virus into the dorsal striatum, we observed transfected SOM somata both in the dorsal striatum (*Figure 1d*, left) and in the cortex (*Figure 1d*, middle left and middle right). These data show that this inhibitory projection is unidirectional, arising preferentially from layer 5 and 6 SOM neurons in the cortex which project to the dorsal striatum.

## Do corticostriatal somatostatin neurons inhibit striatal neurons?

To determine the connectivity pattern of CS-SOM neurons onto neurons in the dorsal striatum, we used an optogenetic approach in which we conditionally expressed channelrhodopsin-2 (ChR2) in SOM neurons by injecting AAV1.ChR2.Flex into the right AC of SOM-Cre transgenic mice (*Figure 2a, b*; injection site *Figure 2c*). After 3–4 weeks, we recorded from the right dorsal striatum, in which ChR2-positive axons could be observed (*Figure 2d*, right). These axons have been reported to remain photoexcitable even when severed from their parent somata (*Petreanu et al., 2007*; *Rock and Apicella, 2015*). To determine the synaptic properties of CS-SOM projections onto striatal neurons, we photoactivated CS-SOM ChR2-positive axons by flashing blue light (470 nm) for 2–5 ms during whole-cell recordings from striatal neurons. IPSCs (*Figure 2f*, red trace) were isolated by applying a command potential of 0 mV (the calculated reversal potential for glutamatergic excitatory conductance). The IPSCs onset of the photo-evoked response was 2.2 ± 0.11 ms (*Figure 2g*, left). This latency is consistent with the IPSCs being the result of a monosynaptic inhibitory input from the cortex and not a local striatal feedback inhibitory network recruited by cortical projections. Blocking excitatory neurotransmission by application of glutamate receptor antagonists NBQX and CPP did not abolish the CS-SOM-ChR2-evoked synaptic IPSCs (*Figure 2f*, magenta trace). In contrast, blocking inhibitory neurotransmission by application of gabazine (*Figure 2f*, black trace) completely abolished the CS-SOM-ChR2-evoked synaptic IPSCs, confirming they were elicited by direct cortical inhibitory transmission. Basic biophysical properties for CS-SOM-ChR2-evoked synaptic IPSCs (*n* = 16) included (*Figure 2g,h,i*): peak, 122 ± 26 pA; charge, 3.85 ± 0.67 pC; rise time, 1.64 ± 0.13 ms; decay time, 28.87 ± 2.76 ms. Biocytin-filled neurons were morphologically identified as SPNs post-hoc by the presence of dendritic spines (*Figure 2e*, right, white arrows). Eight out of 16 neurons were recovered after patching and were processed for imaging; all eight of these neurons showed a high density of dendritic spines at 40-63X magnification. These data reveal that a large proportion of striatal SPNs receive direct inhibitory input driven by CS-SOM neurons but does not exclude the possibility that other striatal neurons also receive inhibitory input from these projections.

To determine how CS-SOM neurons affect the output of striatal neurons, we took advantage of the same viral ChR2 approach described above. We obtained whole-cell recordings from SPNs while injecting a step of current causing the neurons to spike between 2–7 action potentials (*Figure 2j*, black traces). We then photoactivated CS-SOM ChR2-positive axons by flashing blue light (470 nm) for 2–10 ms starting 10–50 ms before the first action potential. Combining current injection with photoactivation of CS-SOM ChR2-positive axons, we observed a delay of the first action potential (*Figure 2j*, blue traces), with an average delay of 159.16 ± 66.45 ms (n=6) (*Figure 2k*). Overall, these

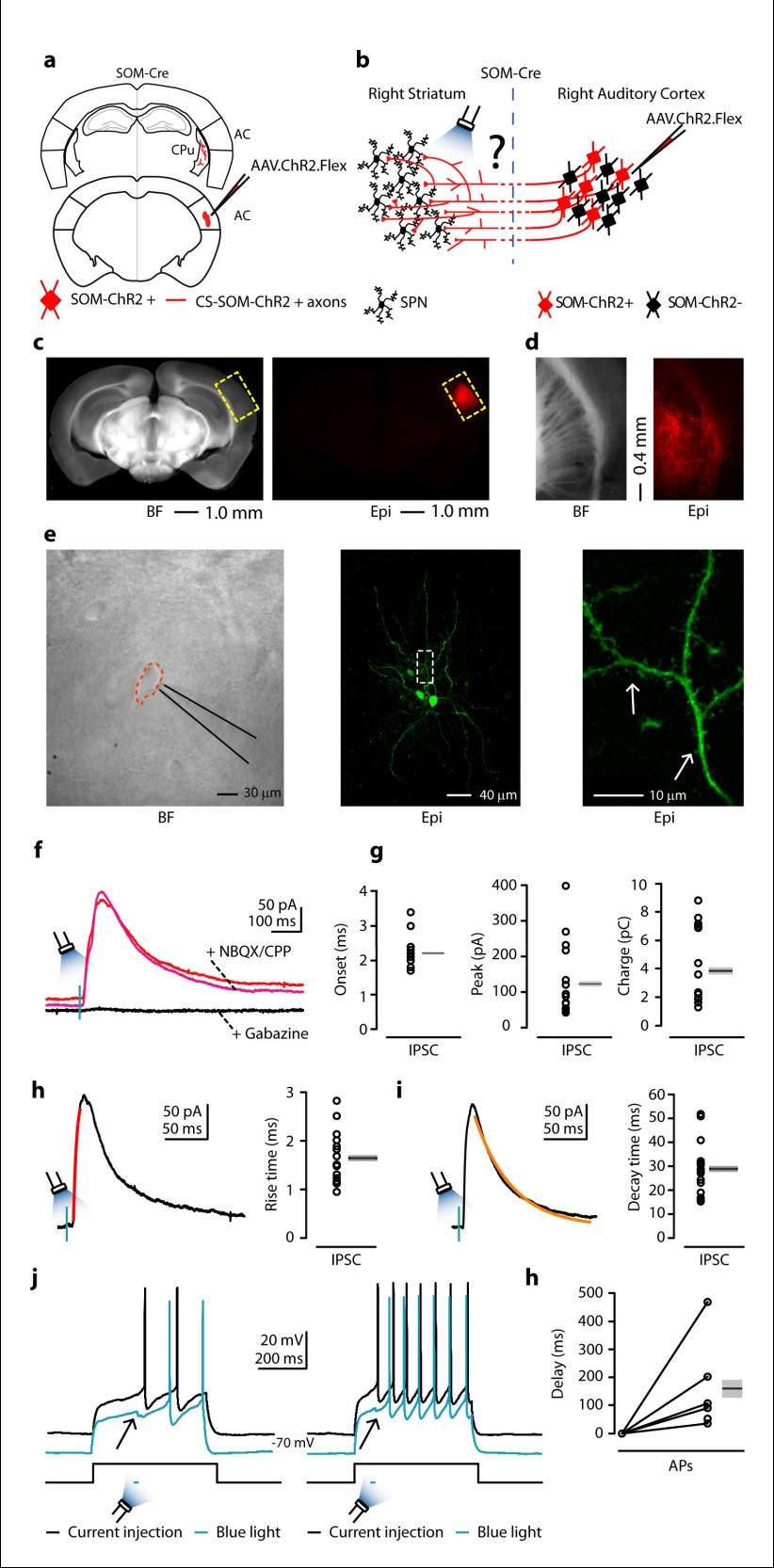

**Figure 2.** Photostimulation of auditory CS-SOM projections elicits direct inhibition and modulates action potentials in striatal SPNs. (a) Schematic depicting injection site using the SOM-Cre transgenic mouse line to

*Figure 2 continued on next page*

*Figure 2 continued*

transfect CS-SOM projections to the dorsal striatum with ChR2. Bottom, auditory cortex: AAV.ChR2.flex injection site. Top, dorsal striatum: red CS-SOM ChR2-tdTomato-positive axons. (**b**) Experimental paradigm for photostimulating ChR2-positive CS-SOM projections while recording from SPNs. (**c**) Bright-field (left) and epifluorescence (right) images of a slice containing the auditory cortex injection site for AAV.ChR2.Flex. (**d**) Bright-field (left) and epifluorescence (right) images of a slice containing the dorsal striatum showing expression of ChR2-tdTomato following injection of AAV.ChR2.Flex into the auditory cortex. (**e**) Left, bright-field image of neurons as seen in bright-field microscopy during patch recordings. Middle, high-resolution epifluorescence image of a biocytin-labeled SPN. The dashed box indicates the location of the image in the right panel. Right, high-resolution epifluorescence image of spines from the biocytin-labeled SPN. (**f**) Example of IPSCs recorded at 0 mV from an SPN before (red trace) and after application of ionotropic glutamate receptor antagonists (NBQX 10 μM, CPP 5 μM: magenta trace) and GABA$_A$ receptor antagonist (gabazine 25 μM: black trace). (**g**) Left, plot of onset latencies recorded in SPNs ($n$ = 16) including group averages (± s.e.m.). Middle, plot of IPSCs peaks calculated for SPNs, including group averages (± s.e.m.). Right, plot of IPSCs charge transfer calculated for individual IPSCs for SPNs, including group averages (± s.e.m.). (**h**) Left, example of IPSCs (black trace) and rising time course (red trace) recorded at 0 mV from an SPN. Right, plot of IPSCs rising time course recorded in SPNs ($n$ = 16) including group averages (± s.e.m.). (**i**) Left, example of IPSCs (black trace) and decay time course (amber trace) recorded at 0 mV from an SPN. Right, plot of IPSCs decay time course recorded in SPNs ($n$ = 16) including group averages (± s.e.m.). (**j**) Left (black trace), response of an SPN in the whole-cell current-clamp configuration to current injection (250 pA, 500 ms; $n$ = 6). Left (blue trace), response of the SPN to current injection with photostimulation of CS-SOM projections (blue bar, 5–20 ms). Right (black trace), response of an SPN in the whole-cell current-clamp configuration to current injection (350 pA, 500 ms; $n$ = 6). Left (blue trace), response of the SPN to current injection with photostimulation of CS-SOM projections (blue bar, 5–20 ms). (**k**) Summary of ChR2-mediated delay of action potential generation in SPNs ($n$ = 6) during current injection combined with photostimulation of the ChR2 CS-SOM projections. Delay was relative to the onset of the first action potential measured during the current injection alone.

results indicate that CS-SOM projections act directly on SPNs and have the ability to affect spike generation and timing in these neurons.

## Cortical inhibition of direct and indirect spiny projection neurons

The dorsal striatum is characterized by two parallel networks: the direct and indirect pathways (*Gerfen et al., 1990*; *Kawaguchi et al., 1990*; *Lei et al., 2004*; *Gerfen and Surmeier, 2011*; *Calabresi et al., 2014*). Direct pathway SPNs (dSPNs) express D1-type dopamine receptors and are suggested to promote behaviors that have previously been rewarded. On the other hand, indirect pathway SPNs (iSPNs) express D2-type dopamine receptors and are suggested to suppress behaviors that have not previously been rewarded (*Gong et al., 2003*; *Ade et al., 2011*; *Calabresi et al., 2014*).

We examined how activity of CS-SOM neurons regulates activity in the direct and indirect pathways. To address this question we used an optogenetic approach in SOM-Cre-D1/D2 transgenic mice (*Figure 3a*) in which dSPNs and iSPNs are labeled with td-Tomato and GFP, respectively (*Figure 3b* left and middle). In these mice, AAV.ChR2.Flex injected in the AC induces ChR2 expression in CS-SOM neurons. Although both D1-type dopamine receptors and our ChR2 virus are tagged with tdTomato, ChR2-tdTomato-positive axons were detectable in the area of our recordings (*Figure 3b*, left and right panels, white arrows). To determine the relative strength of CS-SOM connections onto the two classes of SPNs, we recorded in coronal brain slices from a dSPN and a neighboring iSPN (~100 μm) in the dorsal striatum ipsilateral to the injection site. It has been reported (*Ade et al., 2011*) that in *Drd1a*-tdTomato and *Drd2*-EGFP mice, fluorescent labeling is restricted to SPNs (i.e. no fluorescent labeling of the three main classes of interneurons in the striatum: parvalbumin-expressing, somatostatin-expressing, and cholinergic). However, to ensure that the neurons we recorded from were in fact SPNs, we also identified them post-hoc by the presence of dendritic spines (*Figure 3c*, right, white arrow). Six out of fourteen neurons were recovered after patching and were processed for imaging; all six of these neurons showed a high density of dendritic spines at 40-63X magnification. All recordings were performed in the presence of NBQX and CPP (*Figure 3d*, red traces); CS-SOM-ChR2-evoked synaptic IPSCs were completely blocked by application of gabazine (*Figure 3d*, black traces). The peak and charge of the maximal IPSCs were used as

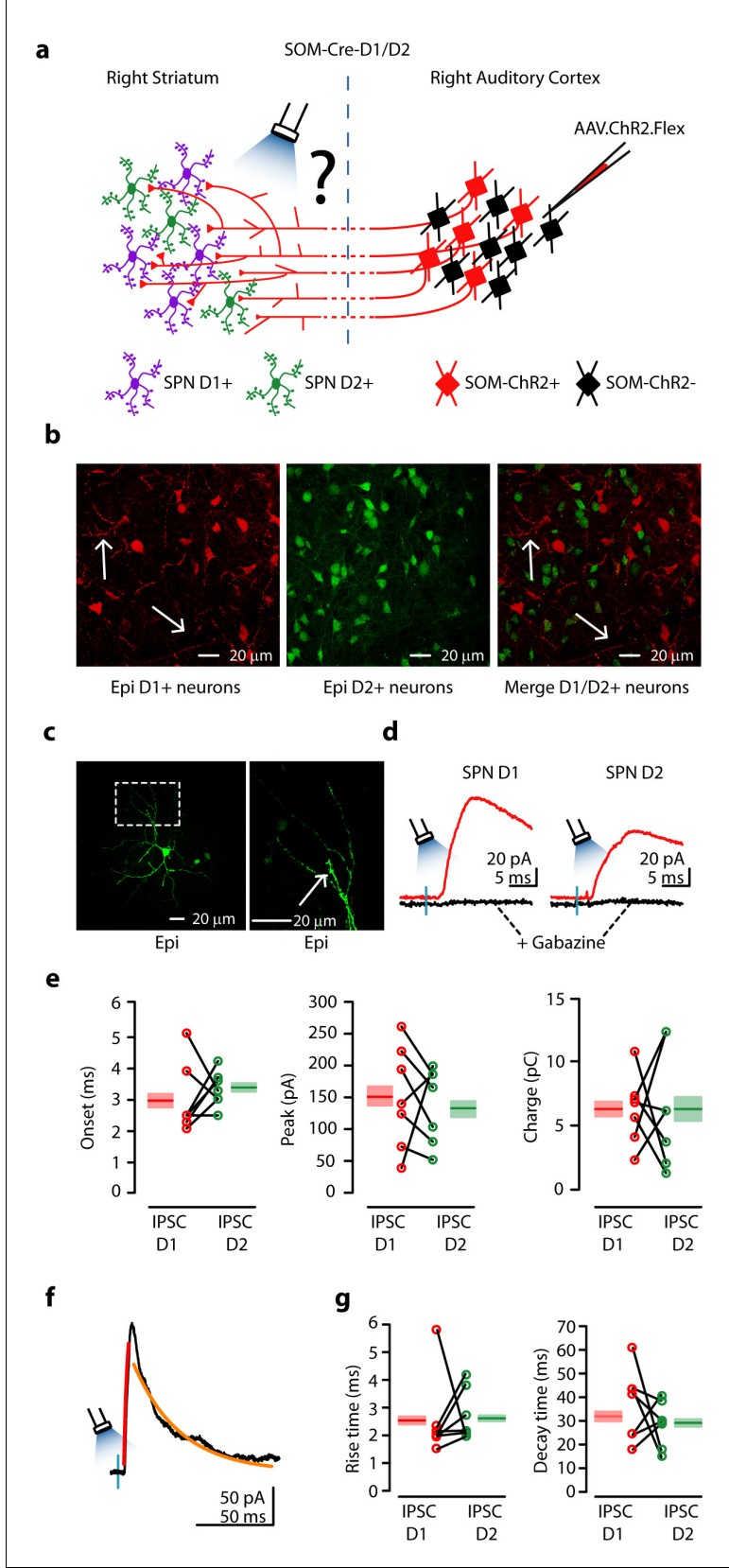

**Figure 3.** Auditory CS-SOM neurons innervate both dSPNs and iSPNs. (**a**) Experimental paradigm for photostimulating ChR2-positive CS-SOM projections while recording from genetically labeled dSPNs and iSPNs.
*Figure 3 continued on next page*

*Figure 3 continued*

(b) Left, epifluorescence image of dSPNs expressing *D1-tdTomato* (white arrows indicate CS-SOM ChR2-tdTomato-positive axons). Middle, epifluorescence image of iSPNs expressing *D2-EGFP*. Right, overlay of *D1-tdTomato* (dSPNs) and *D2-GFP* (iSPNs). Note that the two sub-types of SPNs have a different distribution with no overlap when located in the same region of the dorsal striatum (white arrows indicate CS-SOM ChR2-tdTomato-positive axons). (c) Left, high-resolution epifluorescence image of a biocytin-labeled dSPN. The dashed box indicates the location of the image in right panel. Right, high-resolution epifluorescence image of spines from the biocytin-labeled dSPN. (d) Example of IPSCs recorded at 0 mV from a dSPN (left) and iSPN (right) after application of ionotropic glutamate receptor antagonists (NBQX 10 µM, CPP 5 µM: red traces) and GABA$_A$ receptor antagonist (gabazine 25 µM: black traces). (e) Left, plot of IPSCs onset latencies recorded in dSPNs ($n$ = 7; red circles) and iSPNs ($n$ = 7; green circles), including group averages (± s.e.m.). Middle, plot of IPSCs peaks calculated for dSPNs ($n$ = 7; red circles) and iSPNs ($n$ = 7; green circles), including group averages (± s.e.m.). Right, plot of IPSCs charge calculated for dSPNs ($n$ = 7; red circles) and iSPNs ($n$ = 7; green circles), including group averages (± s.e.m.). (f) Example of IPSCs (black trace), rising time course (red trace) and decay time course (amber trace) recorded at 0 mV from a dSPN. (g) Left, plot of IPSCs rising time course calculated for dSPNs ($n$ = 7; red circles) and iSPNs ($n$ = 7; green circles), including group averages (± s.e.m.). Right, plot of IPSCs decay time course calculated for dSPNs ($n$ = 7; red circles) and iSPNs ($n$ = 7; green circles), including group averages (± s.e.m.).

measures of the strength of the CS-SOM projections to dSPNs (n = 7) and iSPNs (n = 7). IPSCs peak amplitude (dSPN: 149 ± 30 pA; iSPN: 130 ± 21 pA; p = 0.7 rank-sum test) and charge (dSPN: 6.26 ± 1.03 pC; iSPN: 5.24 ± 1.38 pC; p = 0.3 rank-sum test) were similar in dSPNs and iSPNs (*Figure 3e*). There were no differences in IPSCs rise times (*Figure 3g*; example IPSCs from iSPN *Figure 3f*), suggesting the CS-SOM synapses had similar somatodendritic distributions on dSPNs and iSPNs. Together, these results indicate that CS-SOM neurons do not preferentially innervate dSPNs vs. iSPNs in the dorsal striatum.

## A general feature of the GABAergic corticostriatal connectivity pattern

So far we had only observed this projection from the AC to the dorsal striatum. We next asked whether CS-SOM projections from the cortex to the dorsal striatum are a general feature of the corticostriatal circuit organization, or if they are specific to the AC. To test this, we used a similar optogenetic approach in the motor cortex (MC), a cortical area which is not involved in sensory processing like the AC, but instead is involved in the planning, control and execution of voluntary movements.

We used the same methods as in the AC to transfect CS-SOM neurons in the MC with AAV.ChR2.Flex (*Figure 4a,b*; injection site *Figure 4c*). As before, we observed local transfection in the cortex (*Figure 4c*) as well as ChR2-tdTomato-positive axons in the dorsal striatum (*Figure 4d*, white arrows). Neurons in the dorsal striatum where ChR2-tdTomato-positive axons were present were chosen for recordings. We obtained whole-cell recordings from striatal neurons during photoactivation of CS-SOM ChR2-positive axons while recording IPSCs as in the AC (*Figure 4f*, red trace). Responses in this area of the dorsal striatum were similar to those seen in the region of the dorsal striatum receiving auditory CS-SOM projections. Again, application of NBQX and CPP did not abolish the CS-SOM-ChR2-evoked synaptic IPSCs (*Figure 4f*, magenta trace), while application of gabazine completely abolished the CS-SOM-ChR2-evoked synaptic IPSCs (*Figure 4f*, black trace). Basic biophysical properties for CS-SOM-ChR2-evoked synaptic IPSCs ($n$ = 9) included (*Figure 4g,h,i*): onset, 2.4 ± 0.11 ms; peak, 91 ± 17 pA; charge, 3.58 ± 0.59 pC; rise time, 1.76 ± 0.14 ms; decay time, 32.96 ± 4.98 ms. Neurons were identified as SPNs post-hoc by the presence of dendritic spines (*Figure 4e*, right, white arrows). Three out of nine neurons were recovered after patching and were processed for imaging; all three of these neurons showed dendritic spines at 40-63X magnification. Overall, these results indicate that in MC, as well as in AC, CS-SOM projections act directly on SPNs.

## Discussion

In this study we test the hypothesis that the dorsal striatum is influenced by direct GABAergic projections from the cortex. Our results support this hypothesis and additionally conclude that both the motor and the auditory cortices send long-range GABAergic projections to the dorsal striatum, via

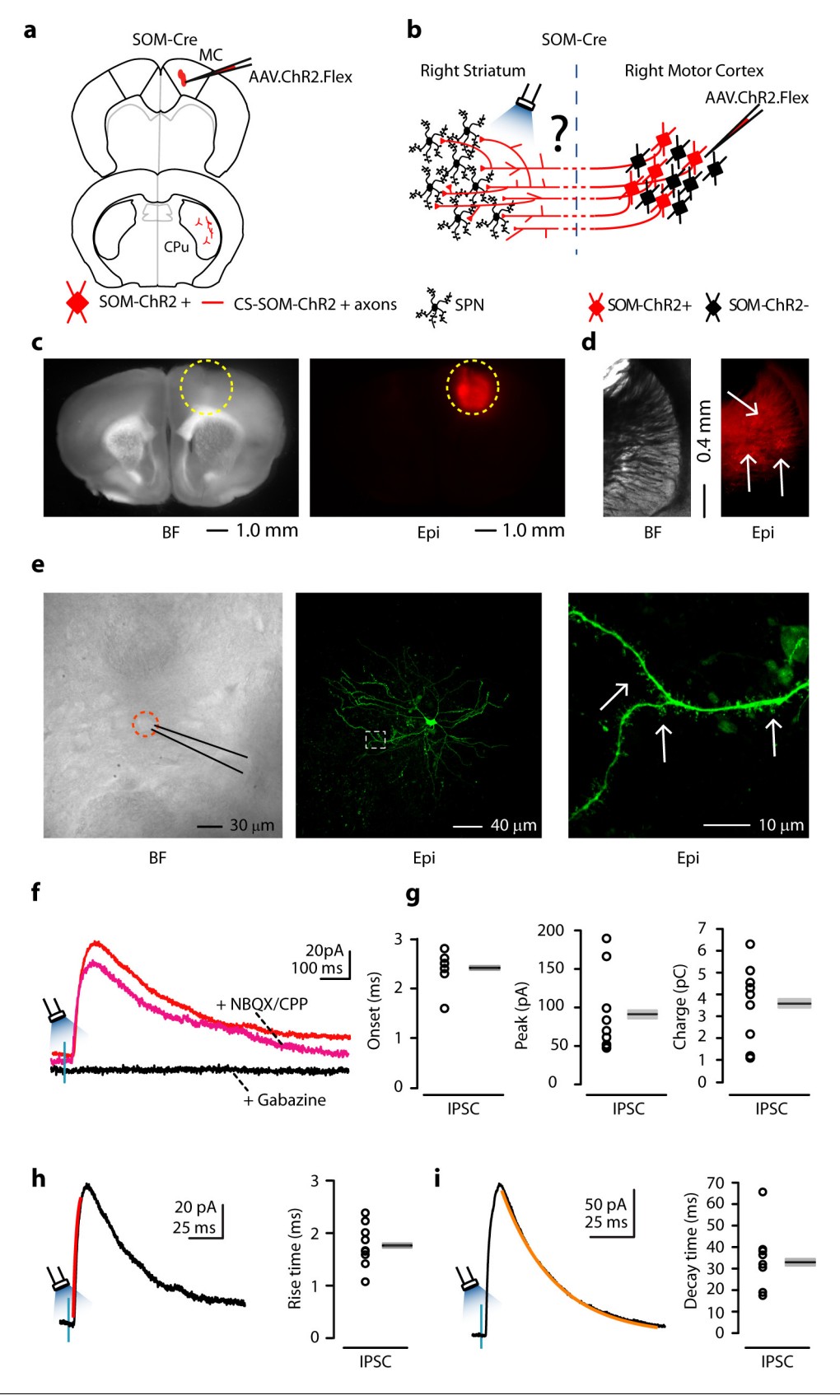

**Figure 4.** Photostimulation of motor CS-SOM projections elicits direct inhibition of striatal SPNs. (a) Schematic depicting injection site using the SOM-Cre transgenic mouse line to transfect CS-SOM projections to the dorsal striatum with ChR2. Top, motor cortex: AAV.ChR2.flex injection site. Bottom, dorsal striatum: red CS-SOM ChR2-tdTomato-positive axons. (b) Experimental paradigm for photostimulating ChR2-positive CS-SOM projections while recording from SPNs. (c) Bright-field (left) and epifluorescence (right) images of a slice containing the motor cortex injection site for AAV.ChR2.Flex. (d) Bright-field (left) and epifluorescence (right) images of a slice containing the dorsal striatum showing expression of ChR2-tdTomato following injection of AAV.ChR2.Flex into the motor cortex. (e) Left, bright-field image of neurons as seen in bright-field microscopy during patch recordings. Middle, high-resolution epifluorescence image of a biocytin-labeled SPN. The dashed box indicates the location of the image in the right panel. Right, high-resolution epifluorescence image of spines from the biocytin-labeled SPN. (f) Example of IPSCs recorded at 0 mV from an SPN before (red trace) and after application of ionotropic glutamate receptor antagonists (NBQX 10 µM, CPP 5 µM: magenta trace) and GABA$_A$ receptor antagonist (gabazine 25 µM: black trace). (g) Left, plot of onset latencies recorded in SPNs ($n$ = 9) including group averages (± s.e.m.). Middle, plot of IPSCs peaks calculated for SPNs, including group averages (± s.e.m.). Right, plot of IPSCs charge transfer calculated for individual IPSCs for SPNs, including group averages (± s.e.m.). (h) Left, example of IPSCs (black trace) and rising time course (red trace) recorded at 0 mV from an SPN. Right, plot of IPSCs rising time course recorded in SPNs ($n$ = 9) including group averages (± s.e.m.). (i) Left, example of IPSCs (black trace) and decay time course (amber trace) recorded at 0 mV from an SPN. Right, plot of IPSCs decay time course recorded in SPNs ($n$ = 9) including group averages (± s.e.m.).

CS-SOM neurons (*Figure 5*). Because of its presence in two such disparate cortical areas, this would suggest that the corticostriatal somatostatin projection connectivity pattern is likely a general feature of the corticostriatal network. We demonstrated that a class of corticostriatal long-range inhibitory neurons in the auditory and motor cortex of the mice are somatostatin-expressing neurons. Our results suggest that CS-SOM neurons provide a major GABAergic projection to the dorsal striatum, in agreement with a recent anatomical study in the mouse frontal cortex (*Tomioka et al., 2015*). However previous results from retrospinal and somatosensory cortex (*Jinno and Kosaka, 2004*) have provided anatomical evidence for expression of parvalbumin in glutamatergic and GABAergic pathways in mice suggesting that corticostriatal long-range inhibitory neurons are characterized predominantly by parvalbumin-expressing neurons. Moreover, recent results from prefrontal cortex (*Lee et al., 2014*), show that corticostriatal long-range inhibitory neurons are characterized predominantly by parvalbumin-expressing or vasoactive intestinal peptide (VIP)-expressing but not somatostatin-expressing interneurons. Overall, these suggest that corticostriatal long-range GABAergic projections may be characterized by heterogeneous subtypes of interneurons, and that additional studies are necessary to determine the subtypes and the physiological impact of long-range corticostriatal GABAergic projections on striatal neurons.

The main finding of the present study is that the cortex can exert a powerful direct inhibition of the dorsal striatum via CS-SOM neurons. Particularly, CS-SOM neurons target SPNs and have the ability to affect their spike timing and generation. In the cortex, interneurons target the somatodendritic compartment of pyramidal neurons (*Somogyi et al., 1998*). SOM cortical and striatal interneurons in particular target the dendrites of pyramidal neurons and SPNs, respectively; although not specifically tuft dendrites as is characteristic of their hippocampal analogue, OLM cells (*Freund and Buzsaki, 1996*; *Kubota and Kawaguchi, 2000*). This connectivity pattern may allow the dendrites to sum incoming activity over a broader window of time (synaptic integration) (*Pouille and Scanziani, 2001*). We suggest that a similar inhibitory mechanism might be applied to the corticostriatal long-range inhibitory circuit, in which CS-SOM neurons, by targeting the dendrites of SPNs, play a role in the integration of incoming cortical excitatory inputs to the striatum.

The dorsal striatum is characterized by two parallel networks: the direct and indirect pathways (*Gerfen et al., 1990*; *Kawaguchi et al., 1990*; *Lei et al., 2004*; *Gerfen and Surmeier, 2011*; *Calabresi et al., 2014*). Particularly, it is suggested that dSPNs promote behaviors that have previously been rewarded while iSPNs suppress behaviors that have not previously been rewarded (*Gong et al., 2003*; *Ade et al., 2011*; *Calabresi et al., 2014*). It is very well established that long-range glutamatergic/excitatory projections regulate the activity of dSPNs and iSPNs. However, how activity of CS-SOM neurons influences dSPNs and iSPNs has not been established. Our results show that the cortex provides an inhibitory projection to both direct and indirect pathway neurons, influencing the output of both striatal networks involved in action selection and suppression of action.

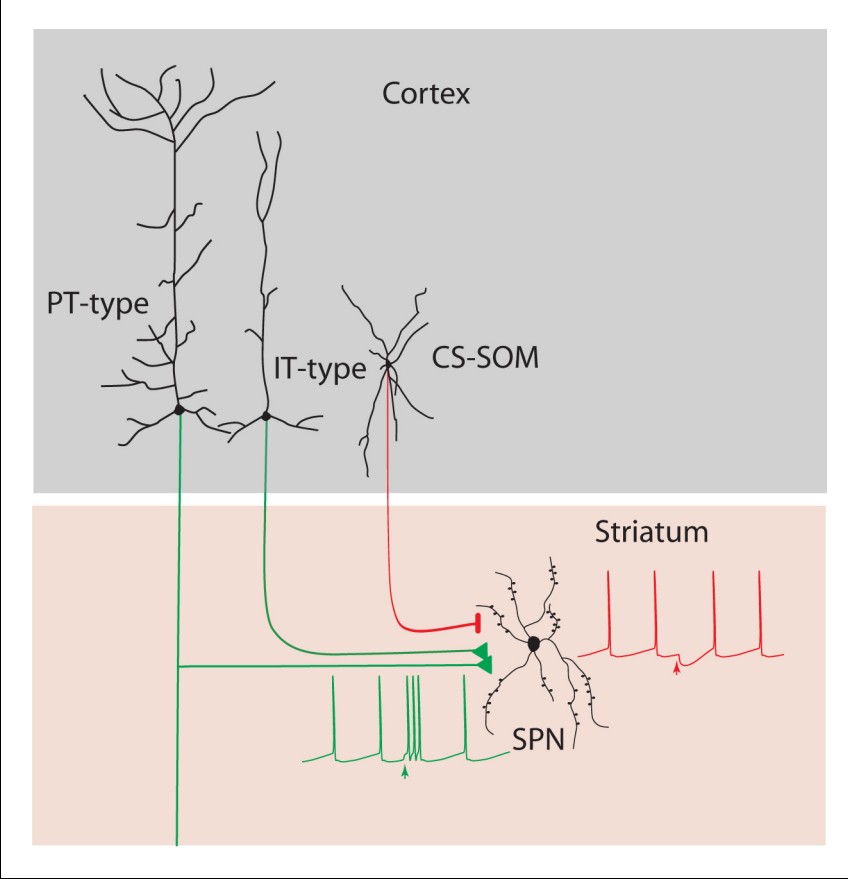

**Figure 5.** Summary diagram: CS-SOM neurons directly inhibit striatal SPNs. Auditory and motor CS-SOM projections modulate the activity of striatal SPNs by direct inhibition. Green lines: excitatory inputs from intratelencephalic (IT-type) and projecting-type (PT-type) layer 5 pyramidal neurons; red line: inhibitory input from CS-SOM neurons.

The functional role of inhibition from CS-SOM neurons might be not in separating the activity of direct and indirect striatal pathways, but rather in shaping the output of both. This is reminiscent of the finding by Kress et al. (*Kress et al., 2013*) that long-range excitatory corticostriatal connections from intratelencephalic and pyramidal tract neurons in motor cortex are made onto both dSPNs and iSPNs in the mouse dorsal striatum.

The dorsal striatum is a brain structure where functionally distinct cortical regions (e.g. motor, auditory, visual, somatosensory) converge (*Wilson et al., 1983*; *Schneider, 1991*; *Flaherty and Graybiel, 1993*; *Chudler et al., 1995*; *Reig and Silberberg, 2014*; *Wilson, 2014*). Anatomical and physiological data have suggested that once this information reaches the dorsal striatum via the corticostriatal projection, it is either channeled into parallel functional segregated circuits (*Alexander et al., 1986*; *Bordi and LeDoux, 1992*; *Bordi et al., 1993*; *Parent and Hazrati, 1995*; *Middleton and Strick, 2000*) or integrated within these circuits (*Nauta and Domesick, 1984*; *Hoffer and Alloway, 2001*; *Kolomiets et al., 2001*; *Reig and Silberberg, 2014*). The existence of direct inhibitory cortical inputs to the dorsal striatum is likely to have profound consequences for corticostriatal mechanisms by which the AC regulates the output of the striatum and how auditory information is transformed into motor command (*Znamenskiy and Zador, 2013*). It is very well established that the dorsal striatum is characterized by activity-dependent plasticity (for review, see [*Kreitzer and Malenka, 2008*]). Particularly, striatal plasticity can alter the transfer of information throughout the striatum and may serve as a neuronal substrate for the transformation of sensory information and motor planning for flexible motor actions and procedural memory. Recently, Xiong et al. (*Xiong et al., 2015*) found that auditory corticostriatal projections are involved in the learning

of an auditory frequency discrimination task. They found that neurons in the dorsal striatum of the rat are selectively potentiated by auditory corticostriatal neurons to promote the learned transformation of sounds into motor actions. Moreover, Chen et al. (*Chen et al., 2015*) demonstrated that motor learning is characterized by structural rearrangements of presynaptic boutons of cortical SOM interneurons. Further experiments are needed to understand the functional significance of this direct cortical inhibition to the dorsal striatum, but in combination with the two above mentioned studies it invites speculation that CS-SOM neurons might play a role in mechanisms underlying spine dynamics by inducing a specific reorganization of dendritic excitatory synapses on SPNs and have consequences on learning and/or memory retrieval.

Our results establish a previously unknown corticostriatal long-range inhibitory circuit (**CS-SOM inhibitory projections → striatal SPNs**) underlying the control of spike timing/generation in striatal spiny neurons and attribute a specific function to a genetically defined type of cortical neuron in corticostriatal communication. We have shown that the dorsal striatum receives not only glutamatergic excitatory inputs from the cortex, but also inhibitory inputs. This may suggest that the timing and ratio of excitation and inhibition, two opposing forces in the mammalian cerebral cortex, can dynamically affect the output of the dorsal striatum, providing a general mechanism for motor control driven by sensory stimuli.

# Materials and methods

All animal procedures were approved by the Institutional Animal Care and Use Committee at the University of Texas at San Antonio. Procedures followed animal welfare guidelines set by the National Institutes of Health. Mice used in this experiment were housed in a vivarium maintaining a 12 hr light/dark schedule and given *ad libidum* access to mouse chow and water.

## Transgenic mouse lines

The following mouse lines were used in this study:

SOM-Cre: Sst*tm2.1(cre)Zjh*/J [The Jackson Laboratory, stock number 013044]; ROSA-tdTomato reporter: B6.CG.Gt(ROSA)26Sor*tm14(CAG-tdTomato)Hze*/J [The Jackson Laboratory, stock number 007914]; *Drd1a*-tdTomato/*Drd2*-GFP reporter: The two reporter lines, B6.Cg-Tg(*Drd1a*-tdTomato) 6Calak/J [The Jackson Laboratory, stock number 016204; (*Ade et al., 2011*)] and Tg(*Drd2*-EGFP) S118Gsat/Mmnc [MMRRC, stock number 000230-UNC] were crossed to generate a reporter for both D1- and D2-expressing neurons. A male mouse positive for both *Drd1a*-tdTomato/*Drd2*-EGFP was provided to us by Dr. Carlos Paladini at UTSA.

SOM-Cre female mice were crossed with a ROSA-tdTomato reporter male mouse to generate a SOM-Cre-tdTomato line (somatostatin-containing neurons expressed both Cre and tdTomato). SOM-Cre female mice were crossed with a *Drd1a*-tdTomato/*Drd2*-GFP reporter male mouse to generate a SOM-Cre-D1/D2 line (somatostatin-containing neurons expressed Cre; D1 neurons expressed tdTomato; D2 neurons expressed GFP).

## Stereotaxic injections

### Basic surgical procedures

Mice were initially anesthetized with isoflurane (3%; 1 L/min O$_2$ flow) in preparation for the stereotaxic injections detailed in the sections below. The mice were head-fixed on a stereotaxic frame (Model 1900; Kopf Instruments) using non-rupture ear bars. Anesthesia was maintained at 1–1.5% isoflurane for the duration of the surgery. A warming pad was used to maintain body temperature during the procedure. Standard aseptic technique was followed for all surgical procedures. Injections were performed using a pressure injector (Nanoject II; Drummond Scientific) mounted on the stereotaxic frame. Injections were delivered through a borosilicate glass injection pipette (Wiretrol II; Drummond Scientific) with a taper length of ~30 mm and a tip diameter of ~50 μm. Both male and female mice, P28-44 at the time of the injection, were utilized in these experiments.

### Anterograde labelling of CS-SOM neurons

CS-SOM neurons in the auditory cortex were labelled using AAV1.CAG.Flex.EGFP.WPRE.bGH (AAV. GFP.Flex; Penn Vector Core) stereotaxically injected into the right auditory cortex of SOM-Cre and

SOM-Cre-tdTomato mice. The injection pipette was positioned over the right auditory cortex (2.5 mm posterior and 4.25–4.35 mm lateral to bregma) and advanced to 0.9–1.2 mm below the surface of the brain. The pipette remained in place for 5 min before the injection began. ~50 nl of AAV. GFP.Flex was delivered over a period of 5–10 min. The pipette was allowed to remain in place for 5 min before being slowly withdrawn.

We injected in four locations in the motor cortex to increase the area of transfection. Stereotaxic coordinates for motor cortex injections: 1.0 mm, 1.2 mm anterior and 1.5 mm, 1.7 mm lateral to bregma; 0.9 mm below the surface of the brain.

### Retrograde labelling of CS-SOM neurons

CS-SOM neurons in the auditory cortex were retrogradely labelled using AAV.GFP.Flex stereotaxically injected into the right auditory dorsal striatum of SOM-Cre mice (3 animals from 2 litters). Injections were performed as above, with the following modifications: Stereotaxic coordinates for the auditory dorsal striatum injection site were 1.45 mm posterior and 3.4–3.5 mm lateral to bregma. ~30–50 nl of AAV.GFP.Flex was delivered between two depths in the dorsal striatum, 2.9 mm and 2.7 mm below the surface of the brain.

### Anterograde transfection of CS-SOM neurons with channelrhodopsin

CS-SOM neurons in the auditory cortex were transfected with channelrhodopsin (ChR2) using AAV1. CAGGS.Flex.ChR2-tdTomato.WPRE.SV40 (AAV.ChR2.Flex; Penn Vector Core) stereotaxically injected into the right auditory cortex of SOM-Cre mice (8 animals from 4 litters) and SOM-Cre-D1/ D2 mice (4 animals from 1 litter). Injections were performed in the same manner as previous injections in the auditory cortex and using the same auditory cortex stereotaxic coordinates.

Motor cortex injections of AAV.ChR2.Flex followed the same basic procedure with motor cortex coordinates as previously mentioned in SOM-Cre mice (6 animals from 4 litters).

### In vitro slice preparation and recordings

We allowed 2–4 weeks for expression of ChR2 or GFP. Mice were anesthetized with isoflurane and decapitated. Coronal slices (300 µm) containing the area of interest (auditory cortex, auditory dorsal striatum, or motor dorsal striatum) were sectioned on a vibratome (VT1200S; Leica) in a chilled cutting solution containing the following (in mM): 100 choline chloride, 25 NaHCO3, 25 D-glucose, 11.6 sodium ascorbate, 7 MgSO4, 3.1 sodium pyruvate, 2.5 KCl, 1.25 NaH2PO4, 0.5 CaCl2. These slices were incubated in oxygenated artificial cerebrospinal fluid (ACSF) in a submerged chamber at 35–37°C for 30 min and then room temperature (21–25°C) until recordings were performed. ACSF contained the following (in mM): 126 NaCl, 26 NaHCO3, 10 D-glucose, 2.5 KCl, 2 CaCl2, 1.25 NaH2PO4, 1 MgCl2; osmolarity was ~290 Osm/L.

Whole-cell recordings were performed in 31–33°C ACSF. Thin-walled borosilicate glass pipettes (Warner Instruments) were pulled on a vertical pipette puller (PC-10; Narishige) and typically were in the range of 3–5 MΩ resistance when filled with a cesium-based intracellular solution, which contained the following (in mM): 110 CsOH, 100 D-gluconic acid, 10 CsCl2, 10 HEPES, 10 phosphocreatine, 1 EGTA, 1 ATP, and 0.3–0.5% biocytin. Inhibitory postsynaptic potentials (IPSCs) were recorded in the voltage-clamp configuration with a holding potential of 0 mV (the calculated reversal potential for glutamatergic excitatory conductances). Intrinsic properties were recorded in the current-clamp configuration using a potassium-based intracellular solution at 31–33°C. Potassium-based intracellular solution contained the following (in mM): 120 potassium gluconate, 20 KCl, 10 HEPES, 10 phosphocreatine, 4 ATP, 0.3 GTP, 0.2 EGTA, and 0.3–0.5% biocytin).

Signals were sampled at 10 kHz and filtered at 4 kHz. Pharmacological blockers used were: CPP (5 µM; Tocris Bioscience), NBQX (10 µM; Abcam), and gabazine (25 µM; Abcam). Hardware control and data acquisition were performed by Ephus (www.ephus.org) (*Suter et al., 2010*).

### ChR2 photostimulation

CS-SOM neurons transfected with ChR2 showed ChR2-tdTomato-positive axons in the dorsal striatum related to the cortical area where the virus was injected (auditory or motor). Because of variability both in ChR2 expression levels (number of ChR2 molecules per transfected neuron) and transfection efficiency (number of ChR2-expressing neurons per animal), to minimize the variability

from experiment to experiment we performed the electrophysiological recording in the same dorsal striatum slices (identified by specific landmarks as slice 1 and 2) and with the highest density of ChR2 transfected axons. We recorded IPSCs from putative spiny projection neurons (SPNs) in the dorsal striatum during photoactivation of the CS-SOM ChR2-positive axon terminals. A 470nm wavelength blue LED (CoolLED *p*E excitation system) passed through a GFP filter cube (Endow GFP/EGFP longpass, C-156625; Chroma) and a 60X water-immersion objective was used to photoactivate CS-SOM ChR2-positive axon terminals.

## Delay/silencing of first action potential in striatal neurons

We recorded from putative SPNs in the dorsal striatum in an area containing ChR2-tdTomato-positive CS-SOM axons. In current-clamp configuration, a step of current was injected to cause the striatal neuron to fire 2–7 action potentials. To determine the effect of CS-SOM projections on the output of striatal neurons, we photoactivated CS-SOM ChR2-positive axons by flashing blue light (470 nm) for 2–10 ms starting 10–50 ms before the first action potential. Combining current injection with photoactivation of CS-SOM projections delayed the current-evoked action potentials in striatal neurons. The action potential delay due to the combined current injection with photoactivation of CS-SOM projections was normalized to the onset of the first action potential measured during the current injection alone.

## Identification of D1- and D2-receptor-expressing striatal neurons

D1- and D2-receptor-expressing neurons in the dorsal striatum were identified by the presence of fluorescent markers in the SOM-Cre-D1/D2 line of mice. Somatic expression of either tdTomato (D1-receptor-expressing neurons) or GFP (D2-receptor-expressing neurons) fluorescence was visualized under a 60X objective.

## Histology

During whole-cell recordings, neurons were filled with an internal solution containing 0.3–0.5% biocytin. Filled neurons were held for at least 20 min, and then the slices were fixed in a formalin solution (neutral buffered, 10% solution; Sigma-Aldrich) for several days at 4°C. The slices were washed well in PBS (6 times, 10 min per wash) and placed in a 4% streptavidin (Alexa Fluor 488, 594, or 680 conjugate; Life Technologies, Carlsbad, CA) solution (498 µl 0.3% Triton X-100 in PBS, 2 µl streptavidin per slice). Slices were allowed to incubate in this solution at 4°C overnight, then washed well in PBS (6 times, 10 min per wash) and mounted with Fluoromount-G (SouthernBiotech) on a glass microscope slide. Confocal images were taken with a Zeiss LSM-710 microscope at varying magnifications (3-63X). The identities of SPNs recorded in the dorsal striatum were confirmed by the presence of spines on their dendritic processes when imaged at 40-63X magnification. Individual high magnification images were stitched together, when necessary, using XuvStitch software (XuvTools). Image adjustment was performed in ImageJ (National Institutes of Health) for brightness/contrast corrections and pseudocoloring.

Some filled neurons were processed for light microscopy. In brief, after overnight incubation in ABC-Elite solution (Vector Laboratories) at 4°C, slices were pre-incubated in 3'3-diaminobenzidine (DAB; Vector Laboratories) for 20 min at 4°C and visualized by adding $H_2O_2$ to the DAB solution. The reaction was stopped when dendritic and axonal processes were visible under light microscopy examination. After washing well in 0.1 M PB (6 times, 10 min per wash), a DAB Enhancing Solution (Vector Laboratories) was applied for 10–20 s to intensify the DAB reaction product in the stained section. Slices were washed again in 0.1 M PB, then mounted with Fluoromount-G on a glass microscope slide for light microscopy. Neurons were morphologically reconstructed in three dimensions using Neurolucida (MicroBrightField) and an upright microscope fitted with a 63X/0.85 oil-immersion objective.

## Data analysis

Figure error bars represent SEM. Data analysis was performed offline using custom MATLAB (MathWorks) routines. Group comparisons were made using the Student's *t*-test if data were normally distributed and the rank-sum test if not, with significance defined as $p < 0.05$.

## Acknowledgements

We are indebted to C Paladini for providing the *Drd1a*-tdTomato/*Drd2*-EGFP transgenic mouse line. We are grateful to G Gaufo for help with confocal imaging; S Lebby for help with single cell staining; S Song for providing SOM-Cre animals.

## Additional information

### Funding

| Funder | Grant reference number | Author |
|---|---|---|
| National Institute of Neurological Disorders and Stroke | NS072197 | Charles Wilson |

The funders had no role in study design, data collection and interpretation, or the decision to submit the work for publication.

### Author contributions

CR, Acquisition of data, Analysis and interpretation of data, Drafting or revising the article, Contributed unpublished essential data or reagents; HZ, Acquisition of data, Analysis and interpretation of data; CW, Analysis and interpretation of data, Drafting or revising the article, Contributed unpublished essential data or reagents; AjA, Conception and design, Acquisition of data, Analysis and interpretation of data, Drafting or revising the article, Contributed unpublished essential data or reagents

### Author ORCIDs

Crystal Rock, http://orcid.org/0000-0002-6986-6166
Alfonso junior Apicella, http://orcid.org/0000-0002-3929-8583

### Ethics

Animal experimentation: Apicella IACUC protocol number: IS00000135 All animal procedures were approved by the Institutional Animal Care and Use Committee at the University of Texas at San Antonio. Procedures followed animal welfare guidelines set by the National Institutes of Health. Mice used in this experiment were housed in a vivarium maintaining a 12 hour light/dark schedule and given ad libidum access to mouse chow and water. Mice were initially anesthetized with isoflurane (3%; 1 L/min O2 flow) in preparation for the stereotaxic injections.

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
