## [Decision Letter]

Thank you for submitting your article "An inhibitory corticostriatal pathway" for consideration by *eLife*. Your article has been favorably evaluated by Gary Westbrook (Senior editor) and three reviewers, one of whom, Sacha Nelson, is a member of our Board of Reviewing Editors.

The following individuals involved in review of your submission have agreed to reveal their identity: Charles Gerfen, Mitsuko Watabe-Uchida (peer reviewers).

The reviewers have discussed the reviews with one another and the Reviewing Editor has drafted this decision to help you prepare a revised submission.

Summary:

The study by Apicella and colleagues characterizes an important recently identified inhibitory cortico-striatal pathway to striatal spiny projection neurons. The significance of this finding cannot be overstated as for over 30 years the assumption has been that cortical input to the striatum is exclusively excitatory. A few partial descriptions have previously indicated an inhibitory component to this project. Here the authors use modern state of the art optogenetic techniques to characterize the nature and physiological impact of this projection.

Essential revisions:

1) The authors should provide better documentation of the neuroanatomy in Figure 1. The main finding in this paper concerns SOM cortical neuron projections to the striatum. The one low magnification of the labeling in the striatum is inadequate, both a low and high magnification image of the distribution of the fibers in the striatum would be nice. Also, the low level images of the retrograde labeling of SOM neurons in the cortex should be added to with higher magnification images that allow one to get a sense of the numbers and laminar distribution of these neurons. It would also be nice if the authors were able to provide a Neurolucida drawing of an axon from one of the cortical neurons going into the striatum (although it is recognized this might be difficult).

2) The rigorous characterization of SOM neurons projecting from layer 5 and 6 of cortex to striatal D1 and D2 MSNs in this paper will be important for the field. However, because the existence of GABAergic projections from the cortex to the striatum, and the fact that some of them were SOM neurons (and PV neurons), have already been reported anatomically and physiologically, it is very important to show what proportion of these GABAergic projections are from SOM neurons and what proportion of MSNs in the striatum receive these projections.

3) The text (Introduction, Results, and Discussion) should more thoroughly discuss past studies of corticostriatal GABAergic projections. For example, are the present physiological results consistent with previous ones in the ventral striatum or different? "A class of GABAergic neurons in the PFC sends long-range projections to NAc and elicits acute avoidance behavior" By Lee et al. in J. Neuroscience 2014 (the most related, including electrophysiology), "Corticofugal GABAergic projection neurons in the mouse frontal cortex" by Tomioka et al. in Frontiers in Neuroanatomy 2015 (mapping of cortical GABAergic projection sites and identification as somatostatin neurons), and "Parvalbumin is expressed in glutamatergic and GABAergic corticostrial pathway in mice." by Jinno and Kosaka in J. Comp. Neurol. 2004 (probably original finding about GABA projection from cortex to striatum). Authors cited them briefly and improperly in the second paragraph of the Introduction and in the first paragraph of the Discussion.

[Editors' note: further revisions were requested prior to acceptance, as described below.]

Thank you for submitting your article "An inhibitory corticostriatal pathway" for consideration by *eLife*. Your article has been favorably evaluated by Gary Westbrook (Senior editor) and three reviewers, one of whom, Sacha B Nelson, is a member of our Board of Reviewing Editors.

The following individuals involved in review of your submission have agreed to reveal their identity: Charles R Gerfen (Reviewer #2); Mitsuko Watabe-Uchida (Reviewer #3).

Although the reviewers felt that most of the points raised were addressed, they did not feel that the slightly higher magnification view of the same image added much to Figure 1 and encourage you to provide an additional image (either as part of Figure 1 or as a supplementary figure) that actually shows greater detail as to the distribution of axons and cell bodies. We will not hold up acceptance of the manuscript on this point if you are unable to provide an additional image in a relatively short period, but all three reviewers felt this would be a significant improvement to the manuscript with relatively little additional effort.

An additional thing that needs to be corrected is the last sentence of the Abstract, "previously unknown" should be deleted as this pathway had been described before.

---

## [Author Response]

Essential revisions:

1) The authors should provide better documentation of the neuroanatomy in Figure 1. The main finding in this paper concerns SOM cortical neuron projections to the striatum. The one low magnification of the labeling in the striatum is inadequate, both a low and high magnification image of the distribution of the fibers in the striatum would be nice. Also, the low level images of the retrograde labeling of SOM neurons in the cortex should be added to with higher magnification images that allow one to get a sense of the numbers and laminar distribution of these neurons. It would also be nice if the authors were able to provide a Neurolucida drawing of an axon from one of the cortical neurons going into the striatum (although it is recognized this might be difficult).

We have added the requested high magnification image of the labeling in the striatum and low magnification image of the retrograde labeling in the cortex to Figure 1 as panel 1B left and 1D middle left and right respectively. We agree with reviewers that would be useful to have a Neurolucida drawing of an axon from one of the cortical neurons going into the striatum. There are too many axons though and they cross over each other too much to allow an axon reconstruction from our virus injections. Of course, intracellular staining of cells in the slice does not fill axons to their terminals in the striatum. It would be very useful to stain individual CS-SOM neurons in vivo using juxtacellular labelling, but so far we have not been successful at this technically challenging experiment.

2) The rigorous characterization of SOM neurons projecting from layer 5 and 6 of cortex to striatal D1 and D2 MSNs in this paper will be important for the field. However, because the existence of GABAergic projections from the cortex to the striatum, and the fact that some of them were SOM neurons (and PV neurons), have already been reported anatomically and physiologically, it is very important to show what proportion of these GABAergic projections are from SOM neurons and what proportion of MSNs in the striatum receive these projections.

In our experiments, *all* the recorded SPNs (MSNs) received inhibitory projections from SOM neurons. Of course, these measurements were made in the region of maximal innervation for our injection sites. Because of variability both in ChR2 expression levels (number of ChR2 molecules per transfected neuron) and transfection efficiency (number of ChR2-expressing neurons per animal), we performed the electrophysiological recording in the same dorsal striatum slices (identified by specific landmarks as slice 1 and 2) and with the highest density of ChR2 transfected axons. We did not see a mosaic structure in the terminal fields, and we think that all SPNs are inhibited by CS-SOM neurons. Of course this does not exclude the possibility that other striatal neurons receive inhibitory input from these projections. We have added this comment in our Results section. The phrase now reads:

“These data reveal that a large proportion of striatal SPNs receive direct inhibitory input driven by CS-SOM neurons but does not exclude the possibility that other striatal neurons also receive inhibitory input from these projections.”

We have now specified in the Materials and methods section a paragraph to clarify how IPSCs recording from SPNs were performed during photo-activation of CS-SOM projections. The paragraph now reads:

“Because of variability both in ChR2 expression levels (number of ChR2 molecules per transfected neuron) and transfection efficiency (number of ChR2-expressing neurons per animal), to minimize the variability from experiment to experiment we performed the electrophysiological recording in the same dorsal striatum slices (identified by specific landmarks as slice 1 and 2) and with the highest density of ChR2 transfected axons.”

We agree with the reviewer that the inhibitory corticostriatal projection probably contains GABAergic cell types in addition to SOM-expressing cells (e.g. PARV-expressing and VIP-expressing cells). We would like to measure the proportion of the total corticostriatal projection accounted for by SOM cells, and also the size of the component arising from other GABAergic neurons. Although it has been shown that PARV-positive neurons project to the striatum (and we have seen this ourselves), we think it is critical to characterize the labelling in the PARV-Cre animal before doing the measurement. Because the pyramidal cell projection to the striatum is so large, even a tiny expression of channelrhodopsin in pyramidal cells would invalidate the measurement. We are currently working on evaluating the selectivity of the PARV Cre mouse and resolving technical problems associated with its use in this experiment.

*3) The text (Introduction, Results, and Discussion) should more thoroughly discuss past studies of corticostriatal GABAergic projections. For example, are the present physiological results consistent with previous ones in the ventral striatum or different? "A class of GABAergic neurons in the PFC sends long-range projections to NAc and elicits acute avoidance behavior" By Lee et al. in J. Neuroscience 2014 (the most related, including electrophysiology), "Corticofugal GABAergic projection neurons in the mouse frontal cortex" by Tomioka et al. in Frontiers in Neuroanatomy 2015 (mapping of cortical GABAergic projection sites and identification as somatostatin neurons), and "Parvalbumin is expressed in glutamatergic and GABAergic corticostrial pathway in mice." by Jinno and Kosaka in J. Comp. Neurol. 2004 (probably original finding about GABA projection from cortex to striatum). Authors cited them briefly and improperly in the second paragraph of the Introduction and in the first paragraph of the Discussion.*

We have made a strenuous effort to provide a deeper analysis of our data in the context of the existing literature (see Discussion). We apologize for our mistaken citation. The citation (Jinno and Kosaka in J. Comp. Neurol. 2004) is properly cited now.

[Editors' note: further revisions were requested prior to acceptance, as described below.]

Although the reviewers felt that most of the points raised were addressed, they did not feel that the slightly higher magnification view of the same image added much to Figure 1 and encourage you to provide an additional image (either as part of Figure 1 or as a supplementary figure) that actually shows greater detail as to the distribution of axons and cell bodies. We will not hold up acceptance of the manuscript on this point if you are unable to provide an additional image in a relatively short period, but all three reviewers felt this would be a significant improvement to the manuscript with relatively little additional effort.

We have added a new higher magnification to the Figure 1 right panel.

*An additional thing that needs to be corrected is the last sentence of the Abstract, "previously unknown" should be deleted as this pathway had been described before.*

We have removed " previously unknown" from the Abstract. The sentence now reads: "Our results describe a corticostriatal long-range inhibitory circuit (CS-SOM inhibitory projections à striatal SPNs) underlying the control of spike timing/generation in SPNs and attributes a specific function to a genetically defined type of cortical interneuron in corticostriatal communication."